# Synthetic, Cell-Derived, Brain-Derived, and Recombinant β-Amyloid: Modelling Alzheimer’s Disease for Research and Drug Development

**DOI:** 10.3390/ijms232315036

**Published:** 2022-11-30

**Authors:** Kseniya B. Varshavskaya, Vladimir A. Mitkevich, Alexander A. Makarov, Evgeny P. Barykin

**Affiliations:** Engelhardt Institute of Molecular Biology, Vavilov St. 32, 119991 Moscow, Russia

**Keywords:** Alzheimer’s disease, beta-amyloid peptide, post-translational modification, amyloidogenesis, NMDAR, tau, LTP, toxic oligomers, amyloid fibrils

## Abstract

Alzheimer’s disease (AD) is the most common cause of dementia in the elderly, characterised by the accumulation of senile plaques and tau tangles, neurodegeneration, and neuroinflammation in the brain. The development of AD is a pathological cascade starting according to the amyloid hypothesis with the accumulation and aggregation of the β-amyloid peptide (Aβ), which induces hyperphosphorylation of tau and promotes the pro-inflammatory activation of microglia leading to synaptic loss and, ultimately, neuronal death. Modelling AD-related processes is important for both studying the molecular basis of the disease and the development of novel therapeutics. The replication of these processes is often achieved with the use of a purified Aβ peptide. However, Aβ preparations obtained from different sources can have strikingly different properties. This review aims to compare the structure and biological effects of Aβ oligomers and aggregates of a higher order: synthetic, recombinant, purified from cell culture, or extracted from brain tissue. The authors summarise the applicability of Aβ preparations for modelling Aβ aggregation, neurotoxicity, cytoskeleton damage, receptor toxicity in vitro and cerebral amyloidosis, synaptic plasticity disruption, and cognitive impairment in vivo and ex vivo. Further, the paper discusses the causes of the reported differences in the effect of Aβ obtained from the sources mentioned above. This review points to the importance of the source of Aβ for AD modelling and could help researchers to choose the optimal way to model the Aβ-induced abnormalities.

## 1. Introduction

Alzheimer’s disease (AD) is a neurodegenerative disorder and the most common cause of dementia [1]. AD is characterised by progressive memory loss and cognitive impairment. A hallmark of AD is the formation of senile plaques in the brain parenchyma, the main component of which is the beta-amyloid peptide (Aβ). Aβ is a fragment of the amyloid precursor protein (APP) produced via sequential proteolysis by β- and γ-secretase enzymes [2]. The imprecise proteolysis by the latter yields peptides 37 to 43 amino acids in length, the most common of which are peptides consisting of 40 and 42 amino acid residues (Aβ40 and Aβ42) [3]. Aβ monomers can form soluble oligomers, which aggregate further into characteristic insoluble fibrils and amyloid plaques [4]. According to the current paradigm, primarily soluble oligomers of Aβ, but also fibrils and plaques, induce neuronal dysfunction and trigger the pathological cascade of AD, including hyperphosphorylation of tau [5,6] and neuroinflammation [7]. Thus, the understanding of the Aβ aggregation process and the biological effects of Aβ assemblies is crucial for the development of diagnostics or disease-modifying therapies for AD. A number of in vitro and in vivo models are used for studying AD pathogenesis and for testing anti-Aβ therapeutics, including systems for monitoring protein aggregation, cell cultures (primary neurons, neuroblastoma lines), acute tissue slices, and animal models (*D. melanogaster, C. elegans*, mice) [8,9,10]. To bridge the gap between traditional 2D culture and in vivo studies, 3D models are being developed, such as brain organoids based on IPSC (induced pluripotent stem cells)-derived human cells [11]. In these models, exogenous Aβ is commonly applied to model the AD-related accumulation of the peptide. Aβ preparations purified from the post-mortem AD brain allow studying the effect of oligomers and fibrils, formed under physiological conditions. However, post-mortem tissue is not easily obtained, and the concentration of soluble oligomers in tissue extracts is low [12,13]. As an alternative, soluble Aβ oligomers are produced in cell cultures overexpressing APP, the most common cell line being Chinese hamster ovarian (CHO) expressing the V717F APP variant (7PA2 cell line). Synthetic Aβ, usually manufactured with solid-phase synthesis (SPS), is widely used in such studies. Despite the structural and functional similarity, Aβ preparations from different sources differ in amyloidogenicity, neurotoxicity, receptor-binding properties, and the ability to activate cellular signalling cascades. Hence, to create a relevant model for studying AD molecular mechanisms or for developing anti-AD drugs, it is needed to choose the most relevant and available source of Aβ. This review discusses the similarities and differences between brain-derived (bAβ), cell culture-derived (secreted, cAβ), synthetic Aβ (sAβ), and recombinant Aβ (rAβ) (Figure 1), with regards to their structure and biological effects, and ponders on the causes of the observed differences.

## 2. Structure of Aβ Aggregates

### 2.1. Synthetic and Brain-Derived Aβ Oligomers and Fibrils

Aβ oligomers can be obtained via the in vitro aggregation of synthetic Aβ or purified from post-mortem tissues (usually, from the brain) of AD patients or transgenic animals expressing human APP. Both synthetic Aβ and brain-derived Aβ can polymerise and form amyloid fibrils [14]. According to Gong et al. [15], Aβ oligomers from these sources have an identical molecular mass and isoelectric point, and are also similar in structure, since they both interact with conformation-specific antibodies. Synthetic Aβ oligomers, like brain-derived Aβ, similarly bind to the brain and hippocampal neurons in primary cultures, which may reflect their structural identity (Figure 2) [15].

However, these Aβ preparations are also significantly different in certain ways. Using sensitivity-enhanced solid-state nuclear magnetic resonance spectroscopy, it was found that fibrils of synthetic Aβ were structurally different, and conformational differences between synthetic, and brain-derived Aβ were observed [16]. sAβ and endogenous Aβ (bAβ and cAβ) species were also different in size and stability [17]: despite similar electrophoretic mobility, they eluted differently in size exclusion chromatography. Unlike bAβ and cAβ, sAβ oligomers can be broken up into monomers if boiled in 1% SDS [17]. Studies with electron and cryo-electron microscopy found that Aβ fibrils obtained in vitro resembled the aggregates purified from the post-mortem human brain (Figure 3). On the other hand, brain-derived fibrils are twisted to the right, in contrast to the left-handed twisting of fibrils formed in vitro [18]. According to the authors of the study, these differences may be not due to the origin of aggregates but to the conditions of fibril formation in vitro. Thus, with careful consideration of the aggregation conditions, it could be possible to obtain in vitro aggregates with a structure similar to natural amyloid fibrils.

### 2.2. Synthetic and Cell-Derived Aβ Aggregates

Studies comparing sAβ and cell-derived Aβ (cAβ) also demonstrated similarities and differences in the properties of these species’ aggregates. Single fibrils of both Aβ species are alike; however, fibrils formed with cAβ purified from primary rat cerebellum cultures demonstrated more complex interfibrillar organisation [19]. Synthetic Aβ aggregation required high concentrations of the peptide (10^−5^–10^−3^ M) and was slower than cAβ aggregation, which started at less than 10^−9^ M [19,20]. sAβ and cAβ showed different mobility in SDS-PAGE, and sAβ aggregation was suppressed by Congo Red, which was not the case for cAβ [21]. Unlike synthetic Aβ, cAβ oligomers, similar to bAβ, were stable in SDS [20,22]. Surprisingly, in the conditioned CHO medium, sAβ oligomerised even in nanomolar concentrations, which points to the possibility that the different properties of these Aβ species could be mitigated with proper conditions for sAβ aggregation [21].

### 2.3. Synthetic and Recombinant Aβ Aggregates

Recombinant expression systems are also used as sources of Aβ. Recombinant Aβ (rAβ) is capable of aggregation and fibril formation; however, rAβ fibrils are longer and less branched than those of sAβ (Figure 4) [23]. Several studies showed that rAβ aggregated faster and with a shorter lag phase than sAβ [23,24].

Thus, brain-derived, cell-derived synthetic and recombinant Aβ can form aggregates but show variation in the aggregation dynamics and fibril structure. Notably, none of the reviewed works studied the aggregation and structure of more than two Aβ species at once, and a thorough comparison of these properties in a single work is yet to be done.

## 3. Biological Effects of Aβ In Vitro and In Vivo

### 3.1. Synthetic and Brain-Derived Aβ Effects in Cultured Cells

Treatment of cultured cells with Aβ is widely used for studying its pathogenic effects and for screening potential anti-Aβ compounds. In such experimental settings, sAβ demonstrates effects, similar to those of the brain-derived peptide. Both sAβ and bAβ induce cell death and neurite degeneration in neuronal cultures [25,26]. Incubation of primary murine hippocampal and cortical neurons with sAβ or with bAβ increases the distance between APP and Presenilin 1 (PSEN1) in the plasma membrane and inhibits Aβ formation via a negative feedback loop [27]. It was also found that sAβ and cAβ alike activated the cAMP response element-binding protein (CREB) via the PI3K/Akt pathway in human neuroblastoma and primary neurons [28]. The following section discusses various aspects of neuronal cell damage in AD that can be modelled with exogenous Aβ treatment and compares the applicability of Aβ from different sources for such studies.

#### 3.1.1. Tau Phosphorylation and Cytoskeletal Disruption

Another hallmark of AD, besides the formation of amyloid plaques, is the hyperphosphorylation and dysfunction of the cytoskeletal protein tau. Tau hyperphosphorylation in the AD brain is likely caused by Aβ and can be observed in cell cultures upon treatment with either synthetic or secreted peptides [29,30,31,32]. Dimers of synthetic and brain-derived Aβ are able to induce hyperphosphorylation of the tau protein in the primary culture of rat hippocampal cells [33]. In a 3D culture model of AD including neurons, astrocytes, and microglia, endogenously produced Aβ at subnanomolar concentrations induced tau phosphorylation, axon damage, and death of neurons and astrocytes [34]. Aβ also detrimentally affects other cytoskeletal proteins: treatment of IPSCs with sAβ oligomers resulted in a decrease in drebrin (actin-binding protein involved in the formation of dendritic spines) [35]. These effects are consistent with the observed loss of drebrin both in the brains of AD patients and in a mouse model of AD [36]. Thus, all three Aβ preparations (sAβ, bAβ, and cAβ) induce tau hyperphosphorylation and destruction of the cytoskeleton. However, subnanomolar (0.2–0.4 nM) concentrations of brain-derived and secreted Aβ were enough to cause such effects, whereas concentrations about 1000 higher (100 nM) were used for sAβ.

#### 3.1.2. Receptor Interaction

Aβ oligomers interact with multiple plasma membrane receptors, such as nicotine acetylcholine receptors, N-methyl D-aspartate receptors (NMDAR), low-density lipoprotein receptor-related protein 1 (LRP1), a receptor for advanced glycation end products (RAGE), and cellular prion protein (PrPc) [37]. The interaction with NMDAR, leading to excitotoxicity, is highly relevant for AD pathology, and the NMDAR antagonist memantine is among the few approved drugs for AD. It was shown that synthetic Aβ bound NMDAR in cultured hippocampal and cortical neurons [38,39]. Activation of NMDAR by sAβ oligomers in micromolar concentrations promotes Ca^2+^ influx [39] and reduces synaptic proteins PSD-95 (postsynaptic density protein 95) and synaptophysin in cultured hippocampal neurons [40], whereas the binding of sub-micromolar (300–500 nM) sAβ to NMDAR induces cellular oxidative stress [41]. Similar effects were observed in primary neurons from mice overexpressing the familial AD variant of human APP–APPswe [42]. In these cells, endogenous cell-secreted Aβ downregulated PSD-95 and reduced cell surface expression of glutamate receptor subunit GluR1, whereas the inhibition of γ-secretase abrogated such changes. Treatment of wild-type neurons with sAβ also affected the levels of PSD-95 and GluR1. The concentration of secreted Aβ was not reported in the study; however, according to other works, cell-secreted Aβ concentration is subnanomolar (about 0.2–0.7 nM in media of SH-SY5Y cells overexpressing APPswe [43]).

Nicotinic acetylcholine receptors (nAChRs) represent another class of receptors mediating the effects of Aβ. In cultured cells, micromolar sAβ exhibits toxicity through nAChR leading to the activation of signalling cascades [44,45]. There are very few works devoted to the interaction of secreted and brain-derived Aβ with nAChR; however, it has been shown that cAβ in the picomolar concentration range changes the morphology of dendritic spines in cell cultures of rat hippocampal neurons, and these effects are prevented by nAChR blockers [46].

In general, both sAβ and cAβ can be used to model AD-related neuronal damage in vitro. However, in all cases considered, to achieve a comparable effect, the required concentrations of the synthetic peptide are 1–2 orders of magnitude higher than those of the cell-derived variant. The effects of recombinant Aβ in cell models are less studied, though it is reported to be more toxic than sAβ, both in cell cultures and in vivo, and may represent a compromise between the availability and the physiological relevance of the effects [24].

#### 3.1.3. Ion Channel Modulation and Neuronal Survival

In the concentration range of pM to low nM, Aβ is thought to exert its physiological effects and support plasticity and neuronal survival [47]. Indeed, if endogenous Aβ is removed from primary neuronal cultures through secretase inhibition or by immunodepletion, neuronal viability is reduced [48], and deregulation of voltage-gated potassium channels occurs [49]. The Aβ production rate or concentration in untransfected cell cultures is rarely reported, though the concentration in cell medium is probably low, in the cerebrospinal fluid of healthy subjects, Aβ40 and Aβ42 are present at about 2 nM and 30 pM, respectively [50,51], and about 1 nM of Aβ42 is found in the water-soluble fraction of human brain extracts [52]. Consistent with these estimations, the addition of 1 nM of synthetic Aβ40 promoted neuronal survival [48] and increased potassium channel expression [49] in Aβ-depleted cultures. Synthetic Aβ_42_ monomers, unlike oligomeric preparations, were shown to be neuroprotective and mitigate the NMDAR excitotoxicity; however, the concentration reported was in a high nanomolar range—30100 nM [53]. However, a study investigating the effect of brain-derived Aβ_42_ found that Aβ_42_ monomers protected primary neurons from Aβ_42_ oligomer neurotoxicity in a much lower concentration of 1 nM [54].

Thus, synthetic Aβ at physiological concentrations (pM–nM) can be robustly used to replicate the role of endogenous cell-secreted peptide in the regulation of neuronal activity and cell viability. Data on the physiological effects of brain-derived Aβ in cell culture are scarce, though it was also reported to be neuroprotective at low concentrations.

### 3.2. Synthetic and Natural Aβ in Animal Models of Alzheimer’s Disease

Studies in animal models have made an important contribution to understanding the pathogenesis of AD. In vivo models can be divided into spontaneous and induced, and the latter can be broken up further into models utilising transgenic and non-transgenic animals [55]. Transgenic models are the most common for AD studies (the transgenes are APP, presenilin 1 (PS-1) and presenilin 2 (PS-2), apolipoprotein E (ApoE), and tau). Despite all the advantages, transgenic animals have a number of character flaws that limit their use in experiments. Transgenic animals model a disease with an early onset, which stands for less than 5% of AD cases. Brain pathology in model animals is characterised by weak activation of microglia and a low level of neuronal death [56,57]. Some authors note that the manifestation of AD pathology after transgene insertion strongly depends on the promoter and genetic background of the animal, which makes it difficult to compare transgenic models with each other [57]. Unlike transgenic animals, induced AD models that are based on the injection of Aβ can cause both neurodegeneration and microglial activation [56]. Injection models allow for the studying of the role of Aβ while controlling its concentration and isoform composition. However, to obtain a relevant model, it is also necessary to choose the optimal source of Aβ. Below, the in vivo use of Aβ obtained from different sources will be compared, which in this review, refers to intracerebroventricular injection (ICV) or intravenous administration of the peptide.

#### 3.2.1. Cerebral Amyloidosis

In contrast to the acute effects of beta-amyloid in the in vitro systems discussed above, cerebral amyloidosis is induced by injection of an exogenous peptide that triggers the misfolding of endogenously generated Aβ peptides. This leads to the development of a pathology that depends on both the host (model animal) and the agent (injected peptide) which was first reported by Meyer-Luehmann et al. [58]. One of the factors complicating studies of amyloidosis induction is that murine Aβ, which differs from human Aβ by three amino acid residues, does not readily aggregate in vivo. Thus, it was shown that the AD brain extract does not induce amyloid pathology in wild-type mice [58,59]. Therefore, transgenic animals expressing human APP variants are predominantly used for this purpose.

To induce cerebral amyloidosis in vivo, brain extracts from both AD patients and transgenic animals overexpressing human APP are used. However, brain-derived Aβ from these sources could have different amyloidogenic activities, which should be taken into consideration. Although both human-derived and mouse-derived Aβ are capable of inducing amyloid pathology [58,60], amyloid plaques in human and transgenic mouse brains have significant differences, such as plaque morphology, solubility and SDS stability, resistance to proteolytic degradation, the repertoire of post-translational modifications [61]. This may be the reason for the difference in the seeding activity of Aβ used for in vivo seeding.

Studies have shown that even within the same species (*Homo sapiens*), there is a heterogeneity in the molecular structure of misfolded Aβ species [62], which facilitated the origin of AD classification, based on the predominant structural type (“strain”) of brain Aβ aggregates. This is consistent with the fact that human-derived Aβ may show variability in its amyloidogenic effects. Thus, it was shown that brain extracts from different AD patients have different seeding activity; although all species induce amyloidosis in mice, different deposition patterns are observed [62,63]. Similarly, the injection of brain extracts obtained from patients with various AD strains induced divergent amyloid pathologies in model animals, different in deposits’ morphology and Aβ modifications [64].

One of the approaches for the search for triggers of cerebral amyloidosis was ICV administration of synthetic Aβ or brain homogenate containing natural Aβ to animals. In one of the studies [65], synthetic Aβ (total of 160 µL 0.5 mg/mL) or human brain homogenate containing native Aβ (total of 300 µL of 10% brain homogenate) was administered over several years to primates (marmosets). Thus, it was found that both synthetic and natural Aβ were able to induce cerebral amyloidosis. Amyloidosis developed in ~90% of monkeys injected with brain homogenate, while in animals inoculated with sAβ, amyloidosis developed in only 10–20% of cases. A similar study on transgenic mice [58] demonstrated that intracerebral injections of 10% brain extract of AD patients over 4 months induced amyloidosis in APP23 transgenic mice; however, the injection of samples containing the synthetic peptide in a similar dose did not cause amyloidosis. The increase in the sAβ dose by 2–3 orders of magnitude led only to the formation of amorphous aggregates near the injection site.

It was also observed that sAβ caused less amyloid deposition compared to the brain-purified sample [66]. Animals inoculated with the synthetic peptide took longer to develop amyloidosis. Since the injected amount of Aβ was approximately 10 times higher in synthetic peptide preparations compared to bAβ preparations, it can be concluded that the amyloidogenic activity of synthetic Aβ is lower [66]. Apparently, a synthetic peptide cannot be considered optimal for the induction of cerebral amyloidosis. More relevant results can be obtained with either amyloid-containing brain extracts or modified forms of the peptide, which are the amyloidogenic components of such extracts [67]. Importantly, the authors of this study failed to find any works studying the amyloidogenic properties of secreted or recombinant Aβ.

#### 3.2.2. Tau Phosphorylation and Cytoskeletal Disruption In Vivo

Modelling of tau pathology in vivo became a major challenge since the pivotal role of this protein in AD pathogenesis became clear [68]. Apart from models combining tau and APP transgenes, models where Aβ is injected in primates or in human tau-expressing mice are widely used to study tau dysfunction. Both synthetic Aβ oligomers and Aβ purified from the human brain or from the brain of APP23 mice are able to induce tau hyperphosphorylation, GSK-3 (glycogen synthase kinase-3) activation, and tau deposition to various degrees. Prolonged (days) injection of 67 ng/h of sAβ oligomers in rTg4510 mice elevated phospho-tau [69], whereas a single 1.5 μL injection of 250 μM of sAβ fibrils in P301L mice induced the formation of neurofibrillary tangles (NFT) [70]. However, similar to the other aspects of AD that were reviewed, sAβ seems to be a less potent trigger of tau pathology than bAβ. In another work on a P301L mice line, a single injection of 2.5 μL of brain extract from APP23 mice containing only 2 nM of Aβ (100–1000 times less than that of sAβ) was enough to cause the tau deposition throughout the P310L mouse brain [71]. In the case of sAβ, the formation of NFT was found in the amygdala only, though it was observed earlier (18 days post-injection) compared to bAβ (6 months post-injection) [70].

#### 3.2.3. Receptor Interaction In Vivo

ICV administration of Aβ preparations is widely used to study the contribution of NMDAR to the pathology of AD. Synthetic Aβ (40–60 pmol) administered to animals inhibited long-term potentiation (LTP) and impaired the cognitive abilities of animals, and these effects were partially abolished by low doses of memantine [72]. Also, using multiphoton imaging, it was shown that picomolar concentrations of cell-secreted Aβ oligomers caused NMDAR-mediated disturbance of calcium homeostasis and synaptic loss in the brain of healthy mice [73]. The results obtained using sAβ and cAβ are consistent with observations in transgenic mice that overexpress Aβ and lack various NMDAR subunits (GluN1, GluN2A, and GluN2B) [74]. Thus, all three Aβ species exert their toxicity via NMDA receptors in vivo and in remarkably low concentrations.

#### 3.2.4. Memory and Cognition

Exogenous peptides are also used to recapitulate cognitive damage, which is a prevalent clinical feature of AD, in animal models. Brain-derived, secreted, and synthetic Aβ peptides upon ICV injection in rats negatively impact cognitive functions [75,76]. However, to observe such effects, different concentrations of peptides are required: micromolar in the case of sAβ and Aβ oligomers purified from transgenic mice brains, and low nM for cAβ. In experiments on arthropods, a cell-secreted form of Aβ caused memory deterioration, which was not the case for synthetic Aβ [19]. Aβ injections can be used to study memory-related forms of synaptic plasticity. Thus, sAβ and Aβ purified from the brain of AD patients caused long-term synaptic depression (LTD) in mice by activating metabotropic glutamate receptor 5 (mGlu5) and binding to the cellular prion protein PrPc [77]. Injection of brain extracts from AD patients [78], Aβ oligomers obtained from both cerebrospinal fluid (CSF) and the 7PA2 cell culture [79], as well as synthetic [80] Aβ, resulted in the inhibition of LTP in vivo.

The influence of Aβ on spatial memory was studied using the Morris water maze and Y-maze on Aβ-injected mice or rats. Multiple studies showed the ability of synthetic Aβ to cause memory impairments in vivo: in some studies, impairments were detected 10–20 days after injection [81,82,83]; however, in other experiments, effects on spatial memory were noticed much later (20–80 days post-injection) [84]. It was also shown that injections of Aβ oligomers caused mild and transient impairment in learning and memory (deterioration was observed on day 12, recovery by day 19) [83]. Such differences can be associated both with variations in the experimental protocols and with the amounts of peptide used, which ranged from 0.135 to 23 μg per animal [81,82,83,84].

Similar to synthetic Aβ, rAβ caused spatial memory impairment in the Morris water maze test (3 μg/μL, 18 μg per animal) 10 days after injection [85]. Notably, human brain-derived Aβ did not impair spatial memory in the Y-Maze and Morris trials (29–39 days post-injection) [86], although social memory impairment was observed (21 days post-injection). Strikingly, the amount of injected Aβ in this study was orders of magnitude lower, less than 3 pg per animal.

The effect of cell-derived Aβ on spatial memory was found using the radial-arm maze: the effect of beta-amyloid oligomers appeared two hours after exposure (∼1–2 nM, 20 μL injection), but disappeared within a day [87]. In a similar study using a radial arm water maze (hybrid of the Morris water maze and radial arm maze), the effect of synthetic Aβ on spatial memory was observed, but over a longer period and using higher concentrations (250 pmol/day infusion of Aβ_42_ within 14 days, total about 16 µg/rat) [88].

Thus, with the exception of brain-derived Aβ, all considered Aβ species (sAβ, rAβ, cAβ) contributed to the impairment of spatial memory, though at different time intervals after injections. The efficacy of different Aβ species in producing memory disruption is difficult to compare directly due to the differences in the experimental protocols. In some works using sAβ and cAβ [83,87], memory recovery was observed, which should be taken into account.

Surprisingly, some studies have shown that the peptides from different sources are analogous when used at low (picomolar) concentrations that exert not toxic, but stimulative effects: in such quantities, both brain-derived and synthetic Aβ enhance LTP in the hippocampus and improve memory in vivo [80,89]. Another experimental work confirmed such findings demonstrating that sAβ oligomers at 200 pM promote the conversion of short-term memory into long-term memory [90]. Thus, brain-derived and synthetic Aβ at picomolar concentrations are both applicable to model the physiological effects of Aβ in vivo, which is in accordance with results obtained in the cell culture.

### 3.3. Studies of Aβ in the Acute Slice Model

Acute hippocampal slices prepared from adult rodents allow studying mature neurons in an almost intact environment under controlled conditions. Though mostly used for measurements of memory-related cellular parameters—LTP and LTD—acute slices were utilised to study the amyloid formation and tau alterations upon application of exogenous Aβ.

#### 3.3.1. Effects of Aβ on LTP and LTD

Aβ oligomers extracted from the cerebral cortex of AD patients are able to inhibit LTP, increase LTD, and reduce the density of dendritic spines in acute slices of the hippocampus [75]. It was also found that Aβ oligomers inhibited LTP regardless of the source (cell cultures, cerebral cortex of AD patients, or synthetic Aβ), and this effect was prevented by NMDAR inhibitors [91]. The contribution of NMDAR to the Aβ effect on neuronal plasticity was shown in another study demonstrating that sAβ, cAβ, and bAβ induced the development of LTD via mGluR or NMDAR [92]. However, differences in the action of these Aβ variants are also reported. Wang et al. [93] showed that although synthetic and cell-derived Aβ both inhibited LTP, cell-derived Aβ was more potent and inhibited LTP at a concentration of 1 nM, while a threshold concentration for sAβ effect was as high as 200 nM or more [72,94,95]. Another study provided an even lower estimation for the concentration of secreted Aβ oligomers (100–300 pM) required to block hippocampal LTP [96]. Thus, peptides obtained in all of the considered ways are able to negatively affect neuronal plasticity in the model of brain slices, though this effect was manifested at different concentrations of Aβ.

On the other hand, similar to in vivo studies, sAβ oligomers in picomolar concentration (200 pM) enhanced the LTP in hippocampal slices via a pathway mediated by α7 nAChR [80,90]. The removal of endogenous Aβ in hippocampal slices with antibodies or small interfering RNA (siRNA) against APP led to a sharp decrease in LTP, which supports the results obtained in vivo [89]. Synthetic Aβ42 at a concentration of 200 pM restored LTP after the endogenous Aβ depletion in slices of wild-type mice [89]. Thus, both synthetic and endogenous Aβ beneficially affect the neuroplasticity within the same picomolar concentration range.

#### 3.3.2. Aβ-Induced Tau Abnormalities in Acute Slices

Most of the studies use synthetic Aβ oligomers to probe the role of tau in AD in acute slices. It was found that sAβ induced LTP block and LTD induction in rodent hippocampal slices via NMDAR-dependent hyperphosphorylation of tau by GSK-3 [97,98]. Oligomers of sAβ also induced tau dysfunction in acute human brain slices [99], and in all of the studies mentioned, the concentration of peptides was in the 100–500 nM range. Interestingly, the same amount of Aβ caused tau hyperphosphorylation and NMDAR toxicity in cell culture (see above). In the only paper on the role of bAβ in acute slices, it was demonstrated that seeding with brain extracts containing ~1 nM of Aβ was required for tau hyperphosphorylation induction by 1.5 μM of sAβ [100].

## 4. Factors behind the Different Properties of Brain-Derived, Cell-Derived, and Synthetic Aβ

Most studies reviewed show that the source of Aβ greatly affects the structure of aggregates of the peptide and its biological effects. However, very few published works are dedicated to a comparison of Aβ obtained from different sources or justify the selection of the Aβ source for the study. Thus, in this review, in many cases, the authors had to compare the results from different studies with different experimental protocols, which complicated the interpretation. The effects of synthetic, secreted, and brain-derived Aβ in various model systems are summarised in Table 1. Recombinant Aβ is used rarely, compared to other variants, and thus was not included in Table 1. Notably, the authors limited the discussion of cell-secreted oligomers that were endogenous for the model since these conditions are difficult to standardise and the concentrations of endogenous oligomers are hardly ever measured in the corresponding works.

Understanding the causes of the different effects of sAβ, cAβ, rAβ, and bAβ is important for both practical and fundamental applications. For sAβ, one of the causes could be the impurity of the peptide. SPS, primarily used for Aβ production, does not yield a product of 100% purity, and the sAβ preparation usually contains trifluoroacetic acid salts and other components that complicate solvation and change the biophysical and biochemical properties of the peptide [102]. Thus, it was shown that different batches of sAβ differed in the rate of fibril formation, toxicity, and binding of Congo Red dye [103,104]. During the synthesis, Met35 oxidation occurs, alongside amino acid racemisation, and both factors hamper the formation of a regular quaternary structure of amyloid aggregates [24]. It has been shown that Met35 oxidation in Aβ42 leads to the formation of oligomers indistinguishable in size and morphology from Aβ40 oligomers [105]. Since Aβ40 oligomers are less toxic than Aβ42 [106], such oxidation can also affect the biochemical properties of synthetic Aβ.

The presence of metal ions in both culture media and in extracellular space in the brain can also stand behind the difference between synthetic and natural Aβ oligomers. Aβ binds divalent cations, of which copper and zinc are considered the most relevant for AD pathology [107]. In the presence of these metals, Aβ aggregates faster and forms aggregates of distinct structures and properties [108], which is prevented by metal chelators [109]. Lipids, such as gangliosides, lipid rafts, cell surface receptors, and other membrane proteins also influence Aβ aggregation through binding to Aβ [110,111]. These components can play the role of a “seed” for pathological Aβ aggregation [111,112]. It was found that the oligomers formed in such conditions differ from synthetic ones in physical properties, such as stability and structure [17,19,21].

Naturally produced Aβ contains multiple variants of different lengths, with either extensions or truncations at the N- and C-termini, whereas synthetic and recombinant peptides of just one specific length (usually 40 or 42 residues) are commonly used. The extended Aβ variants can be toxic, reduce LTP, and impair synaptic functions [113,114]. Truncated Aβ variants are also present in the brains of AD patients and constitute about half of all Aβ molecules [115,116]: such forms are also more toxic and more prone to aggregation [117] than full-length variants and can form stable complexes with full-length Aβ42 [118]. Thus, some proteoforms that contribute to Aβ aggregation and toxicity may not be present in synthetic or recombinant Aβ preparations. From this point of view, it is preferable to use cell-secreted Aβ, since it is close to brain Aβ in terms of N- and C- termini heterogeneity [119].

Truncation is only one example of the Aβ modifications that occur in physiological conditions. In the natural environment, Aβ peptides undergo various post-translational modifications that occur enzymatically or due to the interaction with low molecular weight substances, as well as spontaneously (isomerisation or racemisation) [120]. Such modifications constitute a long list including but not limited to: phosphorylation, isomerisation, citrullination, pyroglutamylation, and covalent cross-linking. Post-translational modifications affect the process of Aβ aggregation, change the structure and the biological effects of the oligomers [121], and influence Aβ hydrolysis by the angiotensin-converting enzyme thereby promoting its accumulation [122]. For example, it was shown that isomerisation of the aspartate residue in the seventh position accelerated the development of amyloidosis in vivo [67] and enhanced the toxic properties of the peptide [123]. The pyroglutamylated Aβ isoform also has an increased propensity for oligomerisation compared to unmodified Aβ [124], as well as resistance to degradation and increased neurotoxicity [125]. Endogenous Aβ oligomers are stabilised with covalent bonds, the formation of which is catalysed either by transglutaminase enzyme or non-enzymatically by prostaglandins and hydroxynonenal; such cross-linked Aβ aggregates are more stable [17]. The formation of highly stable Aβ aggregates can also occur as a result of oxidative modifications, e.g., the formation of dityrosine bonds. Such crosslinks form upon the binding of copper ions, which produce reactive oxygen species in the presence of a reducing agent (for example, ascorbate or dopamine) [126]. Dityrosine-cross-linked Aβ aggregates are found in the brain and cerebrospinal fluid of AD patients [127]. It was shown that the cellular environment also created the necessary conditions for the formation of dityrosine bonds [127], which may explain the higher stability of Aβ oligomers isolated from the brain or obtained from cell cultures compared to synthetic ones. On the contrary, Aβ phosphorylation at Ser8 reduces the aggregation ability of Aβ both in vitro and in a mouse model of AD [128,129].

The divergent effects of modified Aβ species could be related to alterations in the structure of the peptide and of its aggregates. Much of the data on this topic are collected for Aβ of different lengths, primarily comparing 1–40 and 1–42 Aβ [130,131,132,133]. Structural alterations due to isomerisation of Asp residues [134,135], pyroglutamylation [136,137], phosphorylation [138,139], and Met35 oxidation [140,141] were reported as well. The altered chemical and physical properties of modified Aβ assemblies could lead to changes in Aβ receptor binding, as shown for interaction with nicotinic acetylcholine receptors, RAGE, and Na,K-ATPase [128,134,142,143], or to increase in Aβ-related oxidative damage [124,127]. However, the exact mechanisms explaining the enhanced neurotoxicity or aggregation of the modified peptides are mostly unknown.

Nevertheless, various post-translational modifications of Aβ creating a variety of heterogeneous Aβ forms may explain the differences between the properties of synthetic and natural Aβ.

Aside from the variability in the effects of Aβ obtained from different sources, a variance in the effects of different Aβ preparations of the same type should be taken into account. As such, Aβ purification from post-mortem human AD brains or from AD model animals includes the extensive manipulation that most likely alters the native conformation of the aggregates. The conformational variability of aggregates across patients or even brain regions (Aβ strains) was reported [18,62,64]. Synthetic Aβ from different suppliers was shown to have different aggregation kinetics and form aggregates of different structures [144]. Notably, there are multiple protocols for purification of brain-derived Aβ purification [116,145,146,147] and protocols for aggregation of synthetic Aβ [148,149,150] that differ from study to study and from lab to lab. Careful consideration of these factors could improve the quality of the conclusions drawn when comparing the results of different studies.

## 5. Conclusions

The Aβ peptide obtained from different sources—synthetic, cell-secreted, recombinant, and brain-derived—demonstrates significant differences in aggregation properties, in the structure of the oligomers and fibrils, and in the ability to induce biological effects in in vitro and in vivo models. Thus, a synthetic peptide needs to be used in high concentrations to obtain characteristic effects, while a cell-derived peptide is the closest to brain- or CSF-derived Aβ in its effect on cultured cells and acute slices. On the other hand, synthetic and cell-derived peptides are generally more accessible than those isolated from tissues of AD patients or transgenic mice. Hence, the choice of the peptide source should be made based on the experimental model and the studied aspect of AD pathogenesis. Differences in the properties of peptides from different sources can be explained both by conditions for the formation of aggregates (the presence or absence of metal ions, cell membranes receptors, and cross-linking agents) and by a wide range of modified Aβ forms found in an organism or in cell culture. A comparison of the properties of oligomers found in an organism or in cell culture may help to understand the factors that are important for the formation of toxic Aβ aggregates in the AD brain and find novel drug targets or diagnostic markers.

## Figures and Tables

**Figure 1 ijms-23-15036-f001:**
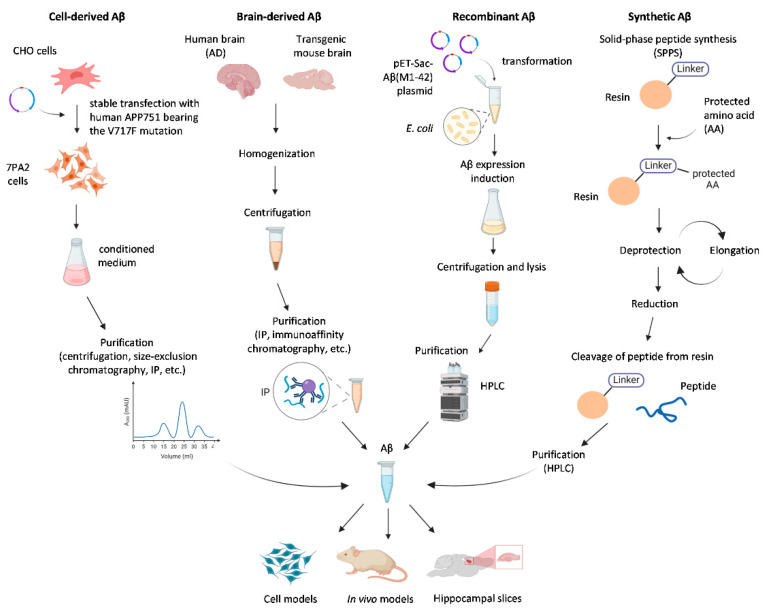
Common sources of beta-amyloid peptide (Aβ) used for modelling Alzheimer’s disease (AD)-related processes in vitro and in vivo.

**Figure 2 ijms-23-15036-f002:**
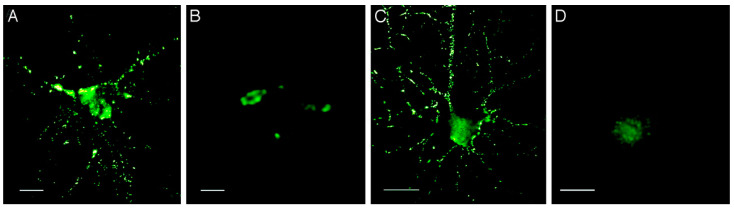
Aβ-derived diffusible ligands (ADDLs) from the AD brain or prepared in vitro show identical punctate binding to neuronal cell-surface proteins. Cultured hippocampal neurons were incubated with soluble extracts of the human brain or synthetic ADDLs. Binding was visualised by immunofluorescence microscopy by using an M93 antibody. Soluble AD-brain proteins (**A**), soluble control-brain proteins (**B**), synthetic ADDLs (**C**), and synthetic ADDLs (**D**) pretreated (1 h) with an oligomer-selective antibody M71 are shown. Small puncta, typically <1 μm, and largely distributed along neurites, are evident for AD extracts and synthetic ADDLs, but not for control extracts or antibody-pretreated ADDLs. (Bar, 10 μm). The figure is reprinted from [15].

**Figure 3 ijms-23-15036-f003:**
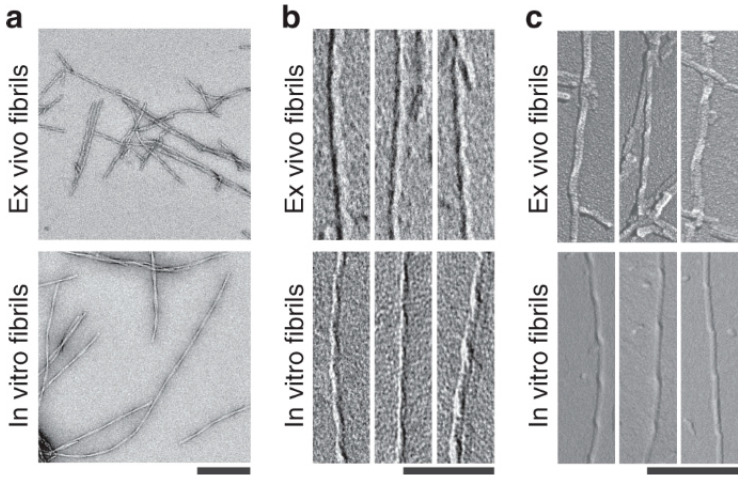
Different structures of brain-derived and in vitro formed Aβ fibrils. (**a**) Negative stain TEM images of brain-derived amyloid fibrils or in vitro formed Aβ fibrils. Scale bar: 200 nm. (**b**,**c**) TEM (**b**) and SEM (**c**) images of brain-derived amyloid fibrils or in vitro formed Aβ fibrils after platinum side shadowing. Scale bars: 100 nm. The figure is reprinted from [18].

**Figure 4 ijms-23-15036-f004:**
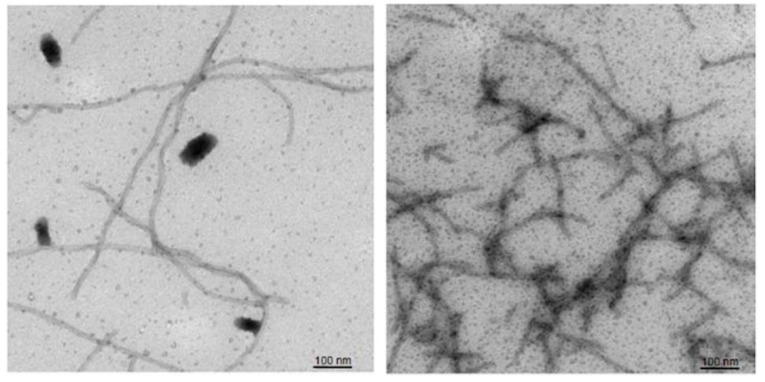
Fibrils of recombinant (**left**) and synthetic (**right**) Aβ42. Negative stain EM analysis. The figure is reprinted from [23].

**Table 1 ijms-23-15036-t001:** Biological effects of synthetic, cell-derived, and brain-derived Aβ in different models.

Effect	Source of Aβ	Models		
		Cells	In Vivo	Slices
Tau phosphorylation and cytoskeletal disruption	Synthetic Aβ	+ (100–500 nM) [31,33,35]	+ (100 µM, single injection or prolonged infusion) [69,70]	+ (100–500 nM) [97,98,99]
	Cell-derived Aβ	++ (0.2–0.5 nM) [29,34]	nd	nd
	Brain-derived Aβ	++ (0.2–0.5 nM) [33]	++ (nM, single injection) [71]	+/− (1 nM, only in combination with sAβ) [100]
Receptor interaction (NMDAR, AMPAR, nAChR)	Synthetic Aβ	+ (300–500 nM–10 µM) [38,39,40,41,44,45]	+ [72]	+ [91,92]
	Cell-derived Aβ	++ (subnanomolar concentrations) [42,46]	++ (pM) [73]	+ [91,92,101]
	Brain-derived Aβ	nd	+ (transgenic animals) [38,74]	+ [91,92]
Cognitive dysfunction/inhibition of LTP	Synthetic Aβ	n/a	+ (µM) [76,77,80]	+ (200–500 nM) [72,91,92,93,94,95]
	Cell-derived Aβ	n/a	++ (pM) [76,79]	++ (100–300 pM–1 nM) [91,92,93,96]
	Brain-derived Aβ	n/a	+ (micromolar concentrations) [75,76,77,78]	+ [75,91,92]
Physiological role: plasticity, survival	Synthetic Aβ	++ (nM) [48,49,53]	++ (pM) [80,90]	++ (pM) [80,90]
	Cell-derived Aβ	+/− (endogenous only) [48,49]	nd	nd
	Brain-derived Aβ	++ (1 nM) [54]	++ (pM, endogenous) [89]	++ (pM, endogenous) [89]

“+”—the presence of an effect, “++”—the presence of an effect at a physiological concentration of Aβ, “+/−”—controversial data, “nd”—no data: no studies were found, “n/a”—not applicable for this model.

## Data Availability

Not applicable.

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
