# Peer review of "Synthetic, Cell-Derived, Brain-Derived, and Recombinant β-Amyloid: Modelling Alzheimer’s Disease for Research and Drug Development"

_ijms, 2022, doi:10.3390/ijms232315036_

Round 1
Reviewer 1 Report
In this review paper, the authors look into the effects of Abeta in vivo and in vitro studies and describe how Abeta species, such as brain-derived Abeta, can influence the result of the studies. This review paper offer not only broad infromation of Abeta studies but also various biological aspects of Abeta. Most of studies just focus on the dark side of Abeta, however, there are several evidence that Abeta exert neuroprotective effects. This review paper includes that kind of information and would be helpful for readers to reconsider Alzheimer's disease pathology.
Author Response
We thank the Reviewer for the high praise of the manuscript.
Reviewer 2 Report
This manuscript deals with a very relevant and understudied topic. Although comparing the different sources of Abeta used within experiments may look trivial and quite technical, the differences described in the literature are quite notorious. In that sense, comparing the different origins of Abeta peptides is of capital importance. Considering this, results must be carefully extrapolated when using different sources of Abeta. This is nicely explained by the authors, and a thorough review of relevant literature is provided. Considering this, the topic developed in this manuscript is relevant. Few issues should be addressed to improve the quality and accuracy of the manuscript:
1. Abstract (lines 13-15) and other parts of the text (e.g. section 1) emphasize on Abeta oligomers. However, most of the experiments summarized in the text deal with fibrils. Considering this, the abstract is misleading and should be rephrased for Abeta aggregates in general. In addition, not only oligomers display the effects explained in lines 13-15 (fibrils also do it although in a lower degree … anyways, oligomers are not the topic of this article).
2. When discussing patient derived Abeta oligomers, several topics needs to be considered. Some of them include the extensive manipulation used to purify them (that most likely alters the native conformation of the aggregates) and the conformational variability of aggregates across patients or even brain regions (Abeta strains). This needs to be included in the manuscript.
3. The sub-section titled “Cerebral amyloidosis” should be further developed for clarification. The prion-like transmission of brain amyloidosis, although induced by exogenously administered misfolded Abeta, is finally developed at expenses of endogenously generated peptides. This needs to be explained, to differentiate from the acute effect induced by Abeta discussed in other sub-sections.
4. Considering the point above, human vs. mouse derived Abeta have also showed differences in their seeding activities. Differences in seeding activity between different AD brains have also been reported. This should be discussed as human derived Abeta may also display variable biological and biochemical activities depending on the donors.
5. The paragraphs dealing with the post-translational modifications on Abeta are quite relevant and needs to be further discussed. It would be nice if the authors speculate on how these modifications could alter the quality (perhaps the structure) of misfolded Abeta, and what mechanisms are responsible for the different effects observed in experimental settings when using them.
